# TOWARDS IMAGE UNDERSTANDING FROM DEEP COMPRESSION WITHOUT DECODING

**Robert Torfason**
ETH Zurich, Merantix
robertto@ethz.ch

**Fabian Mentzer**
ETH Zurich
mentzerf@vision.ee.ethz.ch

**Eirikur Agustsson**
ETH Zurich
aeirikur@vision.ee.ethz.ch

**Michael Tschannen**
ETH Zurich
michaelt@nari.ee.ethz.ch

**Radu Timofte**
ETH Zurich, Merantix
radu.timofte@vision.ee.ethz.ch

**Luc Van Gool**
ETH Zurich, KU Leuven
vangool@vision.ee.ethz.ch

## ABSTRACT

Motivated by recent work on deep neural network (DNN)-based image compression methods showing potential improvements in image quality, savings in storage, and bandwidth reduction, we propose to perform image understanding tasks such as classification and segmentation directly on the compressed representations produced by these compression methods. Since the encoders and decoders in DNN-based compression methods are neural networks with feature-maps as internal representations of the images, we directly integrate these with architectures for image understanding. This bypasses decoding of the compressed representation into RGB space and reduces computational cost. Our study shows that accuracies comparable to networks that operate on compressed RGB images can be achieved while reducing the computational complexity up to $2\times$. Furthermore, we show that synergies are obtained by jointly training compression networks with classification networks on the compressed representations, improving image quality, classification accuracy, and segmentation performance. We find that inference from compressed representations is particularly advantageous compared to inference from compressed RGB images for aggressive compression rates.

## 1 INTRODUCTION

Neural network-based image compression methods have recently emerged as an active area of research. These methods leverage common neural network architectures such as convolutional autoencoders (Ballé et al., 2016; Theis et al., 2017; Rippel & Bourdev, 2017; Agustsson et al., 2017; Li et al., 2017) or recurrent neural networks (Toderici et al., 2015; 2016; Johnston et al., 2017) to compress and reconstruct RGB images, and were shown to outperform JPEG2000 (Taubman & Marcellin, 2001) and even BPG (Bellard) on perceptual metrics such as structural similarity

| Original RGB image | Compressed representation | Decoded RGB image |
|---|---|---|

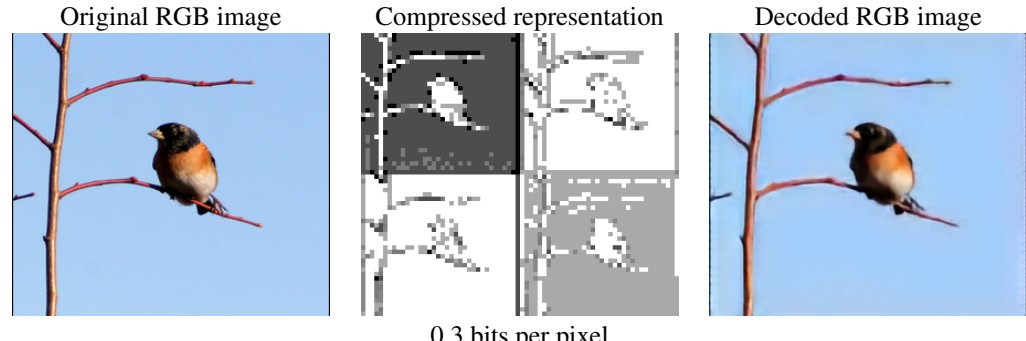

0.3 bits per pixel

Figure 1: We do inference on the learned compressed representation (middle), without decoding.

index (SSIM) (Wang et al. (2004)) and multi-scale structural similarity index (MS-SSIM) (Wang et al. (2003)). In essence, these approaches encode an image $x$ to some feature-map (compressed representation), which is subsequently quantized to a set of symbols $z$. These symbols are then (losslessly) compressed to a bitstream, from which a decoder reconstructs an image $\hat{x}$ of the same dimensions as $x$ (see Fig. 1 and Fig. 2 (a)).

Besides their outstanding compression performance, learned compression algorithms can—in contrast to engineered compression algorithms—easily be adapted to specific target domains such as stereo images, medical images, or aerial images, leading to even better compression rates on the target domain. In this paper, we explore another promising advantage of learned compression algorithms compared to engineered ones, namely the amenability of the compressed representation they produce to learning and inference *without reconstruction* (see Fig. 2). Specifically, instead of reconstructing an RGB image from the (quantized) compressed representation and feeding it to a network for inference (e.g., classification or segmentation), one uses a modified network that bypasses reconstruction of the RGB image.

The rationale behind this approach is that the neural network architectures commonly used for learned compression (in particular the encoders) are similar to the ones commonly used for inference, and learned image encoders are hence, in principle, capable of extracting features relevant for inference tasks. The encoder might learn features relevant for inference purely by training on the compression task, and can be forced to learn these features by training on the compression and inference tasks jointly.

The advantage of learning an encoder for image compression which produces compressed representation containing features relevant for inference is obvious in scenarios where images are transmitted (e.g. from a mobile device) before processing (e.g. in the cloud), as it saves reconstruction of the RGB image as well as part of the feature extraction and hence speeds up processing. A typical use case is a cloud photo storage application where every image is processed immediately upon upload for indexing and search purposes.

Figure 2: We perform inference of some variable $\hat{y}$ from the compressed representation $z$ instead of the decoded RGB $\hat{x}$. The grey blocks denote encoders/decoders of a learned compression network and the white block an inference network.

Our contributions can be summarized as follows:

- We consider two diverse computer vision tasks from compressed image representations, namely image classification and semantic segmentation. Specifically, we use the image compression autoencoder described in (Theis et al., 2017), and adapt ResNet (He et al., 2015) as well as DeepLab (Chen et al., 2016) for inference from the compressed representations.

- We show that image classification from compressed representations is essentially as accurate as from the decompressed images (after re-training on decompressed images), while requiring $1.5 \times - 2 \times$ fewer operations than reconstructing the image and applying the original classifier.

- Further results indicate that semantic segmentation from compressed representations is as accurate as from decompressed images at moderate compression rate, while being more accurate at aggressive compression rates. This suggests that learned compression algorithms might learn semantic features at these aggressive rates or improve localization. Segmentation from compressed representation requires significantly fewer operations than segmentation from decompressed images.

- When jointly training for image compression and classification, we observe an increase in SSIM and MS-SSIM and, at the same time, an improved segmentation and classification accuracy.

- Our method only requires minor changes in the original image compression and classfication/segmentation networks, and slight changes in the corresponding training procedures.

The remainder of the paper is organized as follows. We give an overview over related work in Section 2. In Section 3, we introduce the deep compression architecture we use and in Section 4 we propose a variant of ResNet (He et al., 2015) amenable to compressed representations. We present

and evaluate our methods for image classification and semantic segmentation from compressed representations in Sections 4 and 5, respectively, along with baselines on compressed RGB images. In Section 6, we then address joint training of image compression and classification from compressed representations. Finally, we discuss our findings in Section 7.

## 2  RELATED WORK

In the literature there are a few examples of learning from features extracted from images compressed by *engineered* codecs. Classification of compressed hyperspectral images was studied in (Hahn et al., 2014; Aghagolzadeh & Radha, 2015). Recently, Fu & Guimaraes (2016) proposed an algorithm based on Discrete Cosine Transform (DCT) to compress the images before feeding them to a neural net for reportedly a 2 to $10\times$ speed up of the training with minor image classification accuracy loss. Javed et al. (2017) provide a critical review on document image analysis techniques directly in the compressed domain. To our knowledge, inference from compressed representations produced by *learned* image compression algorithms has not been considered before.

In the context of video analysis, different approaches for inference directly from compressed video (obtained using engineered codecs) were proposed, see (Babu et al., 2016) for an overview. The temporal structure of compressed video streams naturally lends itself to feature extraction for many inference tasks. Examples include video classification (Biswas & Babu, 2013; Chadha et al., 2017) and action recognition (Yeo et al., 2008; Kantorov & Laptev, 2014).

We propose a method that does inference on top of a learned feature representation and hence has a direct relation to unsupervised feature learning using autoencoders. Hinton & Salakhutdinov (2006) proposed a dimensionality reduction scheme using autoencoders to learn robust image features that can be used for classification and regression. A more robust dimensionality reduction was proposed by Vincent et al. (2008) and Rifai et al. (2011) by using denoising autoencoders and by penalizing the Jacobian of the learned representation, respectively, for more robust/stable features. Masci et al. (2011) proposed convolutional autoencoders to learn hierarchical features.

Finally, compression artifacts from both learned and engineered compression algorithms will compromise the performance of inference algorithms. The effect of JPEG compression artifacts on image classification using neural networks was studied in (Dodge & Karam, 2016).

## 3  LEARNED DEEPLY COMPRESSED REPRESENTATION

### 3.1  DEEP COMPRESSION ARCHITECTURE

For image compression, we use the convolutional autoencoder proposed in (Theis et al., 2017) and a variant of the training procedure described in (Agustsson et al., 2017), using scalar quantization. We refer to Appendix A.1 for more details. We note here that the encoder of the convolutional autoencoder produces a compressed representation (feature map) of dimensions $w/8 \times h/8 \times C$, where $w$ and $h$ are the spatial dimensions of the input image, and the number of channels $C$ is a hyperparameter related to the rate $R$. For input RGB images with spatial dimensions $224 \times 224$ the computational complexity of the encoder and the decoder is $3.56 \cdot 10^9$ and $2.85 \cdot 10^9$ FLOPs, respectively.

Quantizing the compressed representation imposes a distortion $D$ on $\hat{x}$ w.r.t. $x$, i.e., it increases the reconstruction error. This is traded for a decrease in entropy of the quantized compressed representation $z$ which leads to a decrease of the length of the bitstream as measured by the rate $R$. Thus, to train the image compression network, we minimize the classical rate-distortion trade-off $D + \beta R$. As a metric for $D$, we use the mean squared error (MSE) between $x$ and $\hat{x}$ and we estimate $R$ using $H(q)$. $H(q)$ is the entropy of the probability distribution over the symbols and is estimated using a histogram of the probability distribution (see (Agustsson et al., 2017) for details). We control the trade-off between MSE and the entropy by adjusting $\beta$. For each $\beta$ we get an operating point where the images have a certain bit rate, as measured by bits per pixel (bpp), and corresponding MSE. To better control the bpp, we introduce the target entropy $H_t$ to formulate our loss:

$$\mathcal{L}_c = \mathrm{MSE}(x, \hat{x}) + \beta \max \left( H(q) - H_t, 0 \right) \tag{1}$$

We train compression networks for three different bpp operating points by adjusting the compression network hyperparameters. We obtain three operating points at 0.0983 bpp, 0.330 bpp and 0.635 bpp[1]. On the ILSVRC2012 data, these operating points outperform JPEG and the newer JPEG-2000 on the perceptual metrics SSIM and MS-SSIM. Appendix A.2 shows plots comparing the operating points to JPEG and JPEG2000 for different similarity metrics and discusses the metrics themselves.

A visualization of the learned compression can be seen in Fig. 1, where we show an RGB-image along with the visualization of the corresponding compressed representation (showing a subset of the channels). For more visualizations of the compressed representations see Appendix A.2.

# 4 IMAGE CLASSIFICATION FROM COMPRESSED REPRESENTATIONS

## 4.1 RESNET FOR RGB IMAGES

For image classification from RGB images we use the ResNet-50 (V1) architecture (He et al., 2015). It is composed of so-called bottleneck residual units where each unit has the same computational cost regardless of the spatial dimension of the input tensor (with the exception of blocks that subsample spatially, and the root-block). The network is fully convolutional and its structure can be seen in Table 1 for inputs with spatial dimension $224 \times 224$.

Following the architectural recipe of He et al. (2015), we adjust the number of 14x14 (conv4_x) blocks to obtain ResNet-71, an intermediate architecture between ResNet-50 and ResNet-101 (see Table 1).

## 4.2 RESNET FOR COMPRESSED REPRESENTATIONS

For input images with spatial dimension $224 \times 224$, the encoder of the compression network outputs a compressed representation with dimensions $28 \times 28 \times C$, where $C$ is the number of channels. We propose a simple variant of the ResNet architecture to use this compressed representation as input. We refer to this variant as **c**ResNet-$k$, where **c** stands for "compressed representation" and $k$ is the number of convolutional layers in the network. These networks are constructed by simply "cutting off" the front of the regular (RGB) ResNet. We simply remove the root-block and the residual layers that have a larger spatial dimension than $28 \times 28$. To adjust the number of layers $k$, we again follow the architectural recipe of He et al. (2015) and only adjust the number of $14 \times 14$ (conv4_x) residual blocks.

Employing this method, we get 3 different architectures: (i) cResNet-39 is ResNet-50 with the first 11 layers removed as described above, significantly reducing computational cost; (ii) cResNet-51 and (iii) cResNet-72 are then obtained by adding $14 \times 14$ residual blocks to match the computational cost of ResNet-50 and ResNet-71, respectively (see last column of Table 1).

A description of these architectures and their computational complexity is given in Table 1 for inputs with spatial dimension $28 \times 28$.

Table 1: Structure of the ResNet and the cResNet architectures in terms of of residual block types, their number, and their associated spatial dimension. Numbers are reported for ResNet-networks with RGB images of spatial dimensions $224 \times 224$ as input, and for cResNet-networks with compressed representations of spatial dimensions $28 \times 28$ as inputs. For a detailed description of the blocks see Appendix A.3

| Network | root | conv2_x $56 \times 56$ | conv3_x $28 \times 28$ | conv4_x $14 \times 14$ | conv5_x $7 \times 7$ | FLOPs $[\times 10^9]$ |
|---|---|---|---|---|---|---|
| ResNet-50 | yes | 3 | 4 | *6* | 3 | 3.86 |
| ResNet-71 | yes | 3 | 4 | *13* | 3 | 5.38 |
| cResNet-39 | no | none | 4 | *6* | 3 | 2.95 |
| cResNet-51 | no | none | 4 | *10* | 3 | 3.83 |
| cResNet-72 | no | none | 4 | *17* | 3 | 5.36 |

---

[1]We obtain the bpp of an operating point by averaging the bpp of all images in the validation set.

### 4.3 Benchmark

We use the ImageNet dataset from the Large Scale Visual Recognition Challenge 2012 (ILSVRC2012) (Russakovsky et al., 2014) to train our image classification networks and our compression network. It consists of 1.28 million training images and 50k validation images. These images are distributed across 1000 diverse classes. For image classification we report top-1 classification accuracy and top-5 classification accuracy on the validation set on $224 \times 224$ center crops for RGB images and $28 \times 28$ center crops for the compressed representation.

### 4.4 Training Procedure

Given a trained compression network, we keep the compression network fixed while training the classification network, both when starting from compressed representations and from reconstructed compressed RGB images. For the compressed representations, we feed the output of the fixed encoder (the compressed representation) as input to the cResNets (decoder is not needed). When training on the reconstructed compressed RGB images, we feed the output of the fixed encoder-decoder (RGB image) to the ResNet. This is done for each operating point reported in Section 3.1.

For training we use the standard hyperparameters and a slightly modified pre-processing procedure from He et al. (2015), described in detail in in Appendix A.4. To speed up training we decay the learning rate at a $3.75\times$ faster speed than in He et al. (2015).

### 4.5 Classification Results

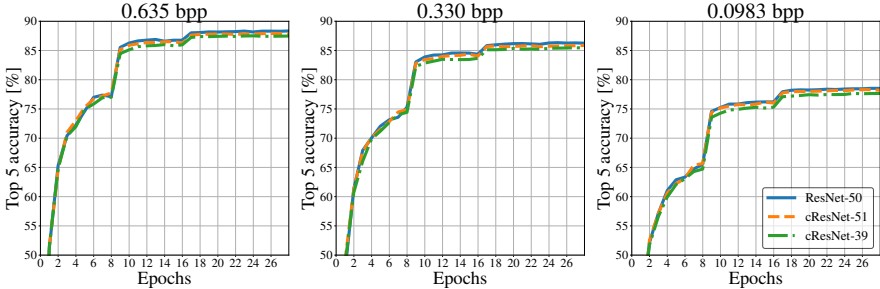

Figure 3: Top-5 accuracy on the validation set for different architectures and input types at each operating point. Results are shown for ResNet-50 (where reconstructed/decoded RGB images are used as input) and for cResNet-51 and cResNet-39 (where compressed representations are used as input).

In Table 2 and in Fig. 3 the results for the classification accuracy of the different architectures at each operating point is listed, both classifying from the compressed representation and the corresponding reconstructed compressed RGB images.

Fig. 3 shows validation curves for ResNet-50, cResNet-51, and cResNet-39. For the 2 classification architectures with the same computational complexity (ResNet-50 and cResNet-51), the validation curves at the 0.635 bpp compression operating point almost coincide, with ResNet-50 performing slightly better. As the rate (bpp) gets smaller this performance gap gets smaller. Table 2 shows the classification results when the different architectures have converged. At the 0.635 bpp operating point, ResNet-50 only performs $0.5\%$ better in top-5 accuracy than cResNet-51, while for the 0.0983 bpp operating point this difference is only $0.3\%$.

Using the same pre-processing and the same learning rate schedule but starting from the original uncompressed RGB images yields $89.96\%$ top-5 accuracy. The top-5 accuracy obtained from the compressed representation at the 0.635 bpp compression operating point, $87.85\%$, is even competitive with that obtained for the original images at a significantly lower storage cost. Specifically, at 0.635 bpp the ImageNet dataset requires 24.8 GB of storage space instead of 144 GB for the original version, a reduction by a factor $5.8\times$.

Table 2: Image classification accuracies after 28 epochs for the $3.75\times$ training rate schedule employed and image segmentation performance for the Deeplab training rate schedule. For each operating point the inputs to ResNet-networks are reconstructed/decoded RGB images and inputs to cResNet-networks are compressed representations. For comparison we show the results with the same training settings, but starting from the original RGB images, in the top row.

| bpp | Network architecture | Top 5 acc. [%] | Top 1 acc. [%] | mIoU [%] |
|---|---|---|---|---|
| | Resnet-50 | 89.96 | 71.06 | 65.75 |
| 0.635 | ResNet-50 | 88.34 | 68.26 | 62.97 |
| | cResNet-51 | 87.85 | 67.68 | 62.86 |
| | cResNet-39 | 87.47 | 67.17 | 61.85 |
| 0.330 | ResNet-50 | 86.25 | 65.18 | 60.75 |
| | cResNet-51 | 85.87 | 64.78 | 61.12 |
| | cResNet-39 | 85.46 | 64.14 | 60.78 |
| 0.0983 | ResNet-50 | 78.52 | 55.30 | 52.97 |
| | cResNet-51 | 78.20 | 55.18 | 54.62 |
| | cResNet-39 | 77.65 | 54.31 | 53.51 |
| | ResNet-71 | 79.28 | 56.23 | 54.55 |
| | cResNet-72 | 79.02 | 55.82 | 55.78 |

To show the computational gains, we plot the top-5 classification accuracy as a function of computational complexity for the 0.0983 bpp compression operating point in Fig. 6. This is done by classification using different architectures that each has an associated computational complexity. The top-5 accuracy of each of these architectures is then plotted as a function of their computational complexity. For the compressed representation we do this for the architectures cResNet-39, cResNet-51 and cResNet-72. For the reconstructed compressed RGB images we used the ResNet-50 and the ResNet-71 architectures.

Looking at a fixed computational cost, the reconstructed compressed RGB images perform about $0.25\%$ better. Looking at a fixed classification cost, inference from the compressed representation costs about $0.6 \cdot 10^9$ FLOPs more. However when accounting for the decoding cost at a fixed classification performance, inference from the reconstructed compressed RGB images costs $2.2 \cdot 10^9$ FLOPs more than inference from the compressed representation.

## 5 SEMANTIC SEGMENTATION FROM COMPRESSED REPRESENTATIONS

### 5.1 DEEP METHOD

For semantic segmentation we use the ResNet-based Deeplab architecture (Chen et al., 2016) and our implementation is adapted using the codes from *DeepLab-ResNet-TensorFlow*[2]. The cResNet and ResNet image classification architectures from Sections 4.1 and 4.2, are re-purposed with atrous convolutions, where the filters are upsampled instead of downsampling the feature maps. This is done to increase their receptive field and to prevent aggressive subsampling of the feature maps, as described in (Chen et al., 2016). For segmentation the ResNet architecture is restructured such that the output feature map has $8\times$ smaller spatial dimension than the original RGB image (instead subsampling by a factor $32\times$ like for classification). When using the cResNets the output feature map has the same spatial dimensions as the input compressed representation (instead of subsampling $4\times$ like for classification). This results in comparable sized feature maps for both the compressed representation and the reconstructed RGB images. Finally the last 1000-way classification layer of these classification architectures is replaced by an atrous spatial pyramid pooling (ASPP) with four parallel branches with rates {6, 12, 18, 24}, which provides the final pixel-wise classification.

---

[2]https://github.com/DrSleep/tensorflow-deeplab-resnet

## 5.2 BENCHMARK

The PASCAL VOC-2012 dataset (Everingham et al. (2015)) for semantic segmentation was used for image segmentation tasks. It has 20 object foreground classes and 1 background class. The dataset consists of 1464 training and 1449 validation images. In every image, each pixel is annotated with one of the 20 + 1 classes. The original dataset is furthermore augmented with extra annotations provided by Hariharan et al. (2011), so the final dataset has 10,582 images for training and 1449 images for validation. All performance is measured on pixelwise intersection-over-union (IoU) averaged over all the classes, or mean-intersection-over-union (mIoU) on the validation set.

## 5.3 TRAINING PROCEDURE

The cResNet/ResNet networks are pre-trained on the ImageNet dataset using the procedure described in Section 4.4 on the image classification task, the encoder and decoder are fixed as in Section 4.4. The architectures are then adapted with dilated convolutions, cResNet-d/ResNet-d, and finetuned on the semantic segmentation task.

For the training of the segmentation architecture we use the same settings as in Chen et al. (2016) with a slightly modified pre-processing procedure as described in Appendix A.5.

## 5.4 SEGMENTATION RESULTS

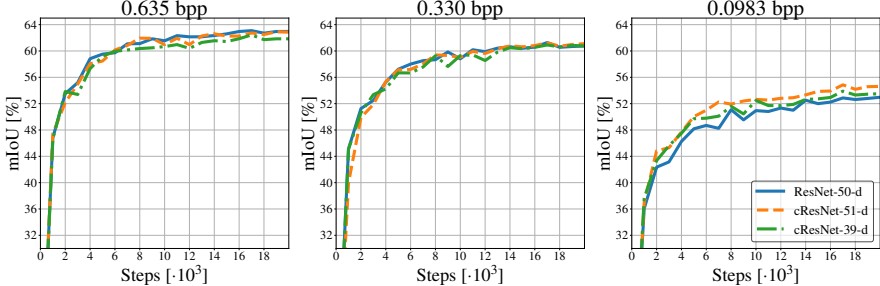

Figure 4: mIoU performance on the validation set for different architectures and input types at each operating point. Results shown for ResNet-50-d (where reconstructed/decoded RGB images are used as input), and for cResNet-51-d and cResNet-39-d (where compressed representations are used as input).

Table 2 and Fig. 4 list the results of the different architectures for semantic segmentation at each operating point, both for segmentation from the compressed representation and the corresponding reconstructed compressed RGB images. Unlike classification, for semantic segmentation ResNet-50-d and cResNet-51-d perform equally well at the 0.635 bpp compression operating point. For the 0.330 bpp operating point, segmentation from the compressed representation performs slightly better, 0.37%, and at the 0.0983 bpp operating point segmentation from the compressed representation performs considerably better than for the reconstructed compressed RGB images, by 1.65%.

Fig. 5 shows the predicted segmentation visually for both the cResNet-51-d and the ResNet-50-d architecture at each operating point. Along with the segmentation it also shows the original uncompressed RGB image and the reconstructed compressed RGB image. These images highlight the challenging nature of these segmentation tasks, but they can nevertheless be performed using the compressed representation. They also clearly indicate that the compression affects the segmentation, as lowering the rate (bpp) progressively removes details in the image. Comparing the segmentation from the reconstructed RGB images to the segmentation from the compressed representation visually, they perform similar. More visual examples are shown in Appendix A.6.

In Fig. 6 we report the mIoU validation performance as a function of computational complexity for the 0.0983 bpp compression operating point. This is done in the same way as in Section 4, using different architectures with different computational complexity, but for segmentation. Here, even without accounting for the decoding cost of the reconstructed images, the compressed representation performs better. At a fixed computational cost, segmentation from the compressed representation

gives about $0.7\%$ better mIoU. And at a fixed mIoU the computational cost is about $3.3 \cdot 10^9$ FLOPs lower for compressed representations. Accounting for the decoding costs this difference becomes $6.1 \cdot 10^9$ FLOPs. due to the nature of the dilated convolutions and the increased feature map size the relative computational gains for segmentation are not as pronounced as for classification.

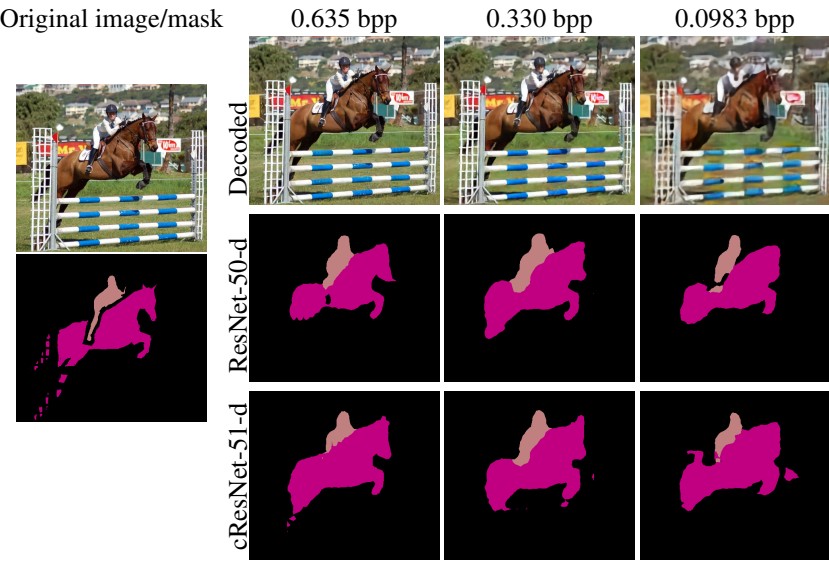

Figure 5: **Top**: Reconstructed/decoded RGB images at different compression operating points. **Middle**: Predicted segmentation mask starting from reconstructed/decoded RGB images using ResNet-50-d architecture. **Bottom**: Predicted segmentation mask starting from compressed representation using cResNet-51-d architecture. **Left**: Original RGB image and the ground truth segmentation mask.

# 6 JOINT TRAINING FOR COMPRESSION AND IMAGE CLASSIFICATION

## 6.1 FORMULATION

To train for compression and classification jointly, we combine the compression network and the cResNet-51 architecture. An overview of the setup can be seen in Fig. 2 b) where all parts, encoder, decoder, and inference network, are trained at the same time. The compressed representation is fed to the decoder to optimize for mean-squared reconstruction error and to a cResNet-51 network to optimize for classification using a cross-entropy loss. The combined loss function takes the form

$$\mathcal{L}_c = \gamma \left( \text{MSE}(x, \hat{x}) + \beta \max \left( H(q) - H_t, 0 \right) \right) + \ell_{ce}(y, \hat{y}), \tag{2}$$

where the loss terms for the compression network, $\text{MSE}(x, \hat{x}) + \beta \max \left( H(q) - H_t, 0 \right)$, are the same as in training for compression only (see Eq. 1). $\ell_{ce}$ is the cross-entropy loss for classification. $\gamma$ controls the trade-off between the compression loss and the classification loss.

When training the cResNet-51 networks for image classification as described in Section 4.4 the compression network is fixed (after having been previously trained as described in Section 3.1). When doing joint training, we first initialize the compression network and the classification network from a trained state obtained as described in Section 3 and 4. After initialization the networks are both finetuned jointly. We initialize from a trained state and our learning rate schedule is short and does not perturb the weights too much from their initial state so we call this finetuning. For a detailed description of hyperparameters used and the training schedule see Appendix A.8.

To control that the change in classification accuracy is not only due to (1) a better compression operating point or (2) the fact that the cResNet is trained longer, we do the following. We obtain a new operating point by finetuning the compression network only using the schedule described above. We then train a cResNet-51 on top of this new operating point from scratch. Finally, keeping

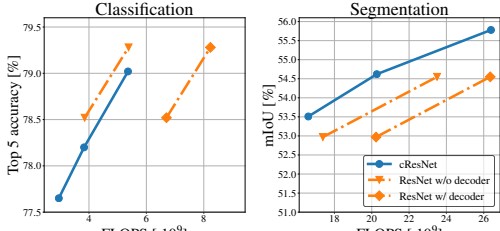 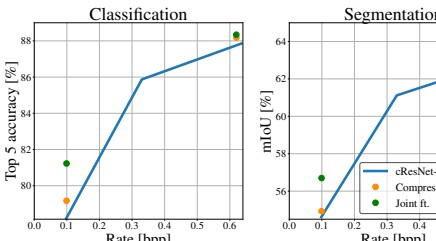

Figure 6: Inference performance at the 0.0983 bpp operating point at different computational complexities, for both compressed representations and RGB images. We report the computational cost of the inference networks only and for reconstructed RGB images we also show the inference cost along with the decoding cost. For runtime benchmarks see Appendix A.9

Figure 7: Showing how classification and segmentation performance improves by finetuning (ft.) the compression network only and the compression network and the classification network jointly. The dots show how the performance "moves up" from the baseline performance when finetuning. The baseline is obtained using fixed compression operating points.

the compression network fixed at the new operating point, we train the cResNet-51 for 9 epochs according to the training schedule above. This procedure controls (1) and (2), and we use it to compare to the joint finetuning.

To obtain segmentation results we take the jointly trained network, fix the compression operating point and adopt the jointly finetuned classification network for segmentation (cResNet-51-d). It is then trained the same way as in Section 5.3. The difference to Section 5.3 is therefore only the pre-trained network.

## 6.2 JOINT TRAINING RESULTS

First, we observe that training the compression and segmentation networks jointly as described in Section 6.1 does not affect the compression performance significantly. In more detail, joint training increases the compression performance on the perceptual metrics MS-SSIM and SSIM by a small amount and decreases the PSNR slightly (high is good for all these metrics), see Appendix A.7.

In Fig. 7 we show how the classification and segmentation metrics change when finetuning the networks (using cResNet-51). It can be seen that the classification and segmentation results "move up" from the baseline through finetuning. By finetuning the compression network only, we get a slight improvement in performance for the classification task but almost no improvements for the segmentation task. However, when training jointly the improvement for classification are larger and we get a significant improvement for segmentation. It is interesting to note that for the 0.635 bpp operating point the classification performance is similar for training the network jointly and training the compression network only, but when using these operating points for segmentation the difference is considerable.

Considering the 0.0983 bpp operating point and looking at the improvements in terms of computational complexity shown in Fig. 6, we see that training the networks jointly, compared to the training only the compression network, we improve classification by 2%, a performance gain which would require an additional 75% of computational complexity of cResNet-51. In a similar way, the segmentation performance after training the networks jointly is 1.7% better in mIoU than training only the compression network. Translating this to computational complexity using Fig. 6, to get this performance by adding layers to the netowork would require an additional 40% of computational complexity of cResNet-51.

## 7 DISCUSSION

We proposed and explored inference when starting directly from learned compressed representations without the need to decode, for two fundamental computer vision tasks: classification and semantic segmentation of images.

In our experiments we departed from a very recent state-of-the-art deep compression architecture proposed by Theis et al. (2017) and showed that the obtained compressed representations can be easily fed to variants of standard state-of-the-art DNN architectures while achieving comparable performance to the unmodified DNN architectures working on the decoded/reconstructed RGB images (see Fig. 6). In particular, only minor changes in the training procedures and hyperparameters of the original compression and classification/segmentation networks were necessary to obtain our results.

The main strong points of the proposed method for image understanding from deep compression without decoding are the following:

**Runtime** Our approach saves decoding time and also DNN inference time as the DNN adapted models can be of smaller depth than those using the decoded RGB images for comparable performance.

**Memory** Removing the need for reconstructing the image is a feat with large potential for real-time memory constrained applications which use specialized hardware such as in the automotive industry. Complementary, we have the benefit of shallower DNN models and aggressive compression rates (low bpp) with good performance.

**Robustness** The approach was successfully validated for image classification and semantic segmentation with minimal changes in the specialized DNN models, which make us to believe that the approach can be extended to most of the related image understanding tasks, such as object detection or structure-from-motion.

**Synergy** The joint training of compression and inference DNN models led to synergistic improvements in both compression quality and classification/segmentation accuracy.

**Performance** According to our experiments and the top performance achieved, compressed representations are a promising alternative to the largely common use of decoded images as starting point in image understanding tasks.

At the same time the approach has a couple of shortcomings:

**Complexity** In comparison with the current standard compression methods (such as JPEG, JPEG2000) the deep encoder we used and the learning process have higher time and memory complexities. However, research on deep compression is in its infancy while techniques such as JPEG are matured. Recently, Rippel & Bourdev (2017) have shown that deep compression algorithms can achieve the same or higher (de)compression speeds as standard compression algorithms on GPUs. As more and more devices are being equipped with dedicated deep learning hardware, deep compression could become commonplace.

**Performance** The proposed approach is particularly suited for aggressive compression rates (low bpp) and wherever the memory constraints and storage are critical. Medium and low bpp compression rates are also the regime where deep compression algorithms considerably outperform standard ones.

Extending our method for learning from compressed representation to other computer vision tasks is an interesting direction for future work. Furthermore, gaining a better understanding of the features/compressed representations learned by image compression networks might lead to interesting applications in the context of unsupervised/semisupervised learning.

## ACKNOWLEDGMENTS

This work was partly supported by ETH Zurich General Fund (OK) and by NVIDIA through a hardware grant.

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

# A  APPENDIX

## A.1  COMPRESSION ARCHITECTURE AND TRAINING PROCEDURE

The compression network is an autoencoder that takes an input image $x$ and outputs $\hat{x}$ as the approximation to the input (see Fig. 2 (a)). The encoder has the following structure: It starts with 2 convolutional layers with spatial subsampling by a factor of 2, followed by 3 residual units, and a final convolutional layer with spatial subsampling by a factor of 2. This results in a $w/8 \times h/8 \times C$-dimensional representation, where $w$ and $h$ are the spatial dimensions of $x$, and the number of channels $C$ is a hyperparameter related to the rate $R$. This representation is then quantized to a discrete set of symbols, forming a compressed representation, $z$.

To get the reconstruction $\hat{x}$, the compressed representation is fed into the decoder, which mirrors the encoder, but uses upsampling and deconvolutions instead of subsampling and convolutions.

To handle the non-differentiability of the quantization step during training, Agustsson et al. (2017) employ a differentiable (soft) approximation of quantization and anneal it to the actual (hard) quantization during training to prevent inversion of the soft quantization approximation. Here, we replace this procedure by a different quantization step, $\bar{Q}$, which behaves like $\hat{Q}$ in the forward pass but like $\tilde{Q}$ in the backward pass (using the notation of (Agustsson et al., 2017)). Note that this is similar to the approach of Theis et al. (2017), who use rounding to integer in forward pass, and the identity function in the backward pass. Like annealing, $\bar{Q}$ prevents inversion of the soft quantization approximation, but facilitates joint training of the autoencoder for image compression with an inference task (see Section 6). Additionally, we chose to use scalar instead of vector quantization (i.e., $p_h = p_w = 1$ in the notation of Agustsson et al. (2017)) to further simplify joint training of compression and inference tasks. This means that each entry of the feature-map is quantized individually.

We train compression networks for three different bpp operating points by choosing different values for $\beta$, $H_t$ and $C$. In theory, changing $H_t$ and $\beta$ is enough to change the resulting average bpp of the network, but we found it beneficial to also change $C$. We obtain three operating points at 0.0983 bpp ($C = 8$), 0.330 bpp ($C = 16$) and 0.635 bpp ($C = 32$)[3]. We use the Adam optimizer (Kingma & Ba, 2014) with learning rates of $1e^{-3}$, $1e^{-5}$, and $1e^{-3}$ for the 0.0983, 0.330 and 0.635 bpp operating points, respectively. We train on the images from the ILSVRC2012 dataset (see Section 4.3), using a batch size of 30. We train each operating point for 600k iterations. Fig. 8 depicts the performance of our deep compression models vs. standard JPEG and JPEG2000 compression on ILSVRC2012 data.

## A.2  IMAGE COMPRESSION METRICS, PERFORMANCE AND VISUALIZATION

We use the following metrics to report performance of our image compression networks: *PSNR* (Peak Signal-to-Noise Ratio) is a standard measure, depending monotonically on mean squared error[4]. *SSIM* (Structural Similarity Index, Wang et al. (2004)) and *MS-SSIM* (Multi-Scale SSIM, Wang et al. (2003)) are metrics proposed to better measure the similarity of images as perceived by humans.

Fig. 8 depicts the performance of our deep compression models vs. standard JPEG and JPEG2000 methods on ILSVRC2012 data on MS-SSIM, SSIM and PSNR. Higher values are always better.

The compressed representations learned in the compression network are visualized in Fig. 9. The original RGB-image is shown along with compressed versions of the RGB image which reconstructed from the compressed representation. In the interest of space we only visualize 4 channels of the compressed representation for each image, even though each operating point has more than 4 channels. We choose the 4 channels with the highest entropy. These visualizations indicate how the networks compress an image, as the rate (bpp) gets lower the entropy cost of the network forces the compressed representation to use fewer quantization centers, as can clearly be seen in Fig. 9. For the most aggressive compression, the channel maps use only 2 centers for the compressed representation.

---

[3]We obtain the bpp of an operating point by averaging the bpp of all images in the validation set.

[4] $\text{PSNR} = 10 \cdot \log_{10}\left(255^2/\text{MSE}\right)$

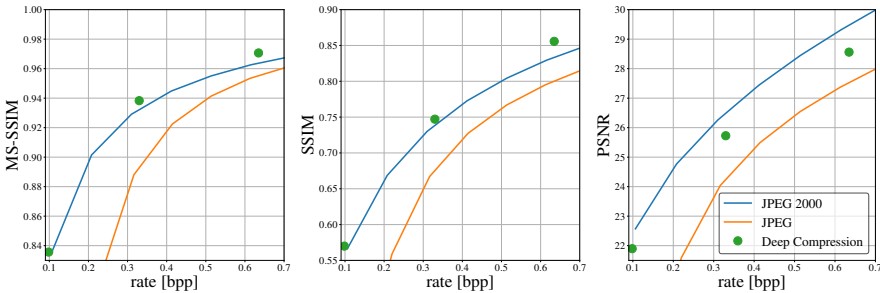

Figure 8: MS-SSIM, SSIM and PSNR as a function of rate in bpp. Shown for JPEG 2000, JPEG and the reported Deep Compression operating points. Higher is better.

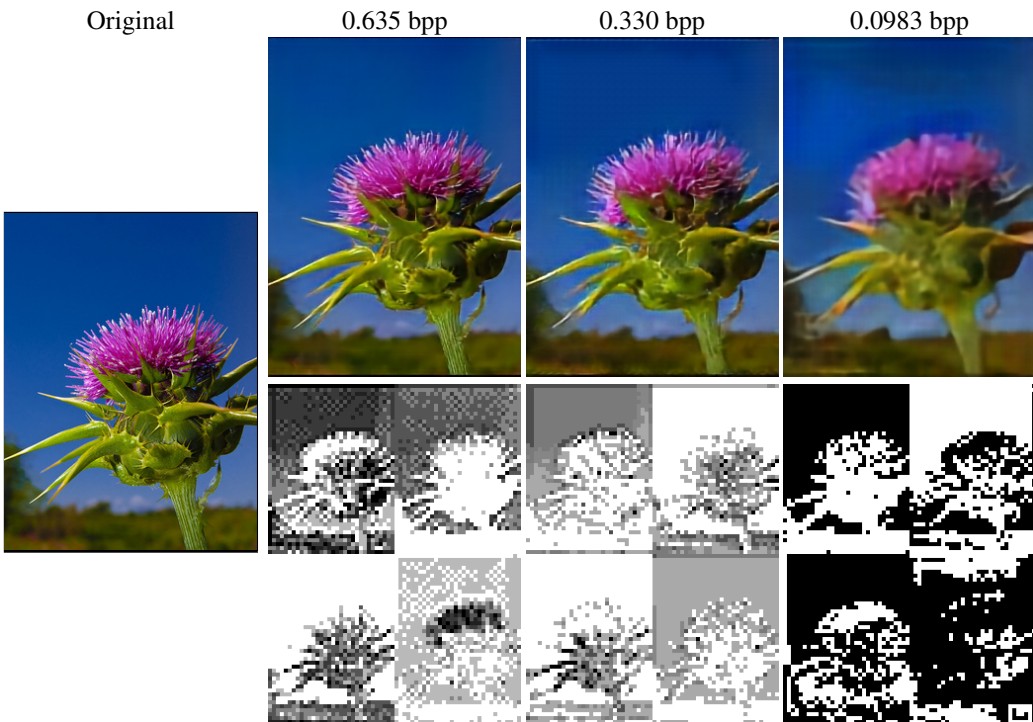

Figure 9: For each operating point we show the reconstructed/decoded image along with the 4 highest entropy channels of the compressed representation. The original RGB image is shown on the left for comparison. The channels of the compressed representation look like quantized downscaled versions of the original image, which motivates doing inference based on them instead of the reconstructed RGB images.

## A.3 Architecture Table

Table 3 is a more detailed version of Table 1 and shows the detailed structure of the networks used, with the dimensions of the convolutions inside the network shown along with all layers.

| layer name | output size | RGB | | | | Compressed representation | | | | | |
|---|---|---|---|---|---|---|---|---|---|---|---|
| | | ResNet-71 | | ResNet-50 | | cResNet-72 | | cResNet-51 | | cResNet-39 | |
| conv2_x | 56×56 | 3×3 max pool, stride 2 | | | | | | | | | |
| | | 1×1, 64
3×3, 64
1×1, 256 | ×3 | 1×1, 64
3×3, 64
1×1, 256 | ×3 | None | | None | | None | |
| conv3_x | 28×28 | 1×1, 128
3×3, 128
1×1, 512 | ×4 | 1×1, 128
3×3, 128
1×1, 512 | ×4 | 1×1, 128
3×3, 128
1×1, 512 | ×4 | 1×1, 128
3×3, 128
1×1, 512 | ×4 | 1×1, 128
3×3, 128
1×1, 512 | ×4 |
| conv4_x | 14×14 | 1×1, 256
3×3, 256
1×1, 1024 | ×**13** | 1×1, 256
3×3, 256
1×1, 1024 | ×**6** | 1×1, 256
3×3, 256
1×1, 1024 | ×**17** | 1×1, 256
3×3, 256
1×1, 1024 | ×**10** | 1×1, 256
3×3, 256
1×1, 1024 | ×**6** |
| conv5_x | 7×7 | 1×1, 512
3×3, 512
1×1, 2048 | ×3 | 1×1, 512
3×3, 512
1×1, 2048 | ×3 | 1×1, 512
3×3, 512
1×1, 2048 | ×3 | 1×1, 512
3×3, 512
1×1, 2048 | ×3 | 1×1, 512
3×3, 512
1×1, 2048 | ×3 |
| | 1×1 | average pool, 1000-d fc, softmax | | | | | | | | | |
| FLOPs | | $5.38\times10^9$ | | $3.86\times10^9$ | | $5.36\times10^9$ | | $3.83\times10^9$ | | $2.95\times10^9$ | |

Table 3: Structure of the ResNet and the cResNet architectures. The numbers reported are for ResNet-networks where the inputs are RGB images with a spatial dimensions $224 \times 224$ and for cResNet-networks where the inputs are compressed representations with spatial dimensions $28 \times 28$. Building blocks are shown in brackets, with the numbers of blocks stacked. Downsampling is performed by conv3_1, conv4_1, and conv5_1 with a stride of 2.

## A.4 Training Classification

We use the ResNet implementation from the Slim library in TensorFlow[5] with modifications for the custom architectures. For a fair comparison when using different settings we train all classifications networks from scratch in our experiments. For the training we use a batch size 64 and employ the linear scaling rule from Goyal et al. (2017) and use the learning rate 0.025. We employ the same learning rate schedule as in (He et al., 2015), but for faster training iterations we decay the learning rate $3.75\times$ faster. We use a constant learning rate that is divided by a factor of 10 at 8, 16, and 24 epochs and we train for a total of 28 epochs.

A stochastic gradient descent (SGD) optimizer is used with momentum 0.9. We use weight decay of 0.0001. For pre-processing we do random-mirroring of inputs, random-cropping of inputs ($224\times224$ for RGB images, $28 \times 28$ for compressed representations) and center the images using per channel mean over the ImageNet dataset.

## A.5 Training Segmentation

For the training of the segmentation architecture we use the same settings as in Chen et al. (2016) with a slightly modified pre-processing procedure. We use batch size 10 and perform 20k iterations for training using SGD optimizer with momentum 0.9. The initial learning rate is 0.001 (0.01 for final classification layer) and the learning rate policy is as follows: at each step the initial learning rate is multiplied by $(1 - \frac{\text{iter}}{\text{max\_iter}})^{0.9}$. We use a weight decay of 0.0005. For preprocessing we do random-mirroring of inputs, random-cropping of inputs ($320 \times 320$ for RGB images, $40 \times 40$ for the compressed representation) and center the images using per channel mean over the dataset.

## A.6 Segmentation Visualization

Fig. 10 shows visual results of segmentation from compressed representation and reconstructed RGB images as in Fig. 5. The performance is visually similar for all operating points except for the 0.0983 bpp operating point in Fig. 10 where the reconstructed RGB image fails to capture the back part of the train, while the compressed representation manages to capture that aspect of the image in the segmentation.

---

[5] https://www.tensorflow.org

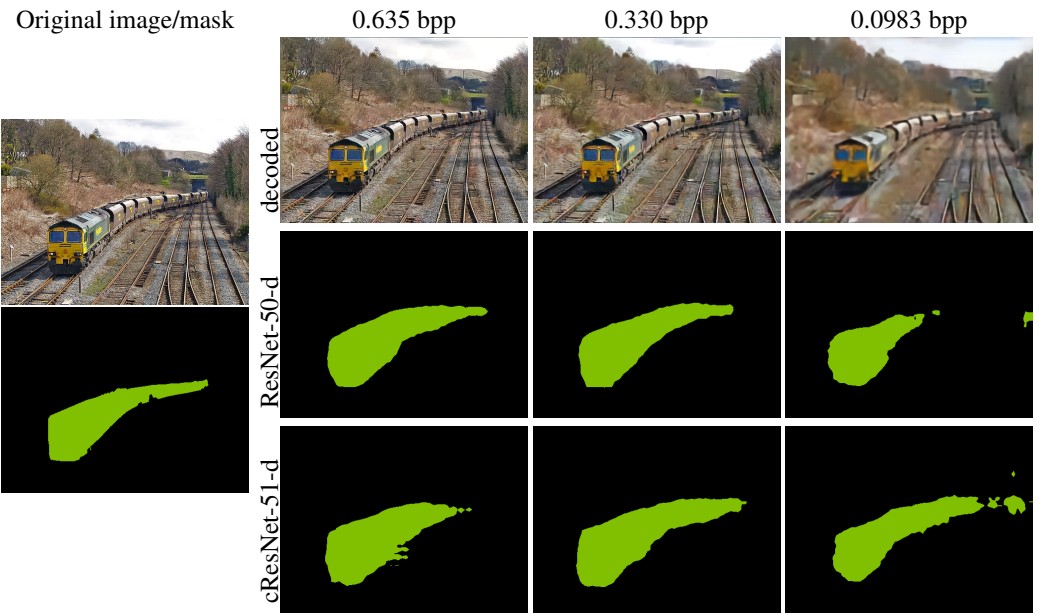

Figure 10: **Top**: Reconstructed/decoded RGB images at different compression operating points. **Middle**: Predicted segmentation mask starting from reconstructed/decoded RGB images using ResNet-50-d architecture. **Bottom**: Predicted segmentation mask starting from compressed representation using cResNet-51-d architecture. **Left**: Original RGB image and the ground truth segmentation mask.

## A.7    IMAGE COMPRESSION METRICS FOR JOINT TRAINING

In Fig. 11 the compression metric results of finetuning the whole joint network (joint ft.), are compared to finetuning only the compression network (compression ft.). In both cases the same learning rate schedule is used, namely the one described in Section 6.1 Each image shows 4 distinct points along with a baseline for JPEG-2000. These 4 points are:

- *joint-1*: The joint ft. operating point at the beginning of the finetuning

- *joint-2*: The joint ft. operating point at the end of finetuning

- *compression-1*: the compression ft. operating point at the beginning of the finetuning

- *compression-2*: the compression ft. operating point at the end of finetuning

*joint-1* and *compression-1* are the same because both joint ft. and compression ft. are initialized from the same starting point. An arrow then shows how the operating point for the joint training moves from *joint-1* to *joint-2* after finetuning. In the same manner an arrow shows how point *compression-1* moves to *compression-2* after finetuning.

Fig. 11 shows how the points move in the rate-vs.-{MSSSIM,SSIM,PSNR} plane. When training, hitting an exact target bpp is difficult due to the noisy nature of the entropy loss. Therefore the points in Fig. 11 do not only move along the y-axis (MS-SSIM, SSIM or PSNR) but also move along the x-axis (rate). We show the final results for both joint ft. and compression ft. at the same bpp for a fair comparison.

As is evident from Fig. 11 this finetuning procedure improves the image compression metrics in all cases, i.e., they converge at a higher value for a lower bpp. We re-iterate, for all metrics higher values are better. However for SSIM and MS-SSIM the joint ft. improves *more* than the compression ft. For PSNR, however, the joint ft. improves less than the compression ft. The same effect is consistent for both 0.0983 and the 0.635 bpp operating points.

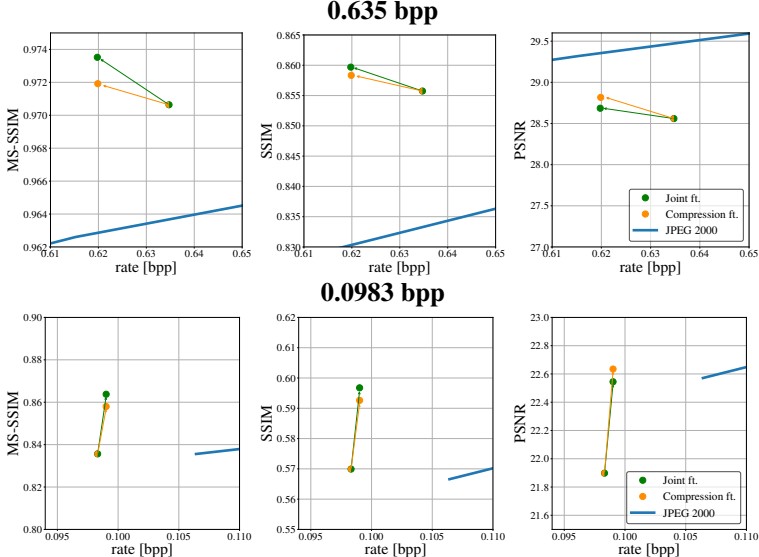

Figure 11: Showing how the selected metrics move from the original compression operating point to a different point after finetuning. We show this change when finetuning the compression network only, and then when finetuning the compression network and the classification architecture jointly. **Top**: 0.635 bpp operating point **Bottom**: 0.0983 bpp operating point.

## A.8 Joint Training Training and Hyperparameters

For joint training we set the hyperparameters in Eq. 2 to $\gamma = 0.001$, $\beta = 150$ and $H_t = 1.265$ for the 0.635 bpp operating point and $\gamma = 0.001$, $\beta = 600$ and $H_t = 0.8$ for the 0.0983 bpp operating point.

The learning rate schedule is similar to the one used in the image classification setting. It starts with an initial learning rate of 0.0025 that is divided by 10 every 3 epochs using a SGD optimizer with momentum 0.9. The joint network is then trained for a total of 9 epochs.

## A.9 Runtimes benchmarks

In Fig. 12 we show the average runtimes (per image) for different setups. This complements Fig. 6 where we showed the theoretical computations for each setup. All benchmarks were run on a GeForce Titan X GPU in TensorFlow v1.3. We used batch size 256 for classification and batch size 20 for segmentation. For RGB images we used the image spatial dimension $224 \times 224$ and for the compressed representations we used spatial dimension $28 \times 28$ (corresponding to a $224 \times 224$ input image to the compression network).

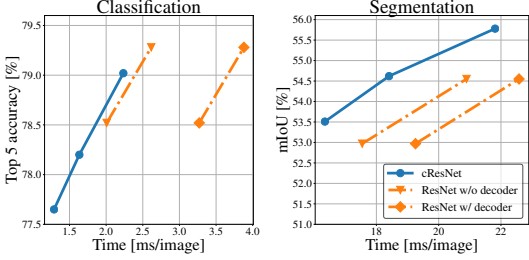

Figure 12: Inference performance at the 0.0983 bpp operating point for different architectures, for both compressed representations and reconstructed RGB images. We report the computational runtime (per image) of the inference networks only and for the reconstructed RGB images we also show the runtime for the inference network along with the decoding runtime.

