# OpenReview forum: "Towards Image Understanding from Deep Compression Without Decoding"
_ICLR.cc/2018/Conference — Accept (Poster)_

### Official Review · AnonReviewer2 · 2017-11-25
**obvious but welcomed application for neural-net based image compression**

**Rating:** 9
**Confidence:** 5

**Review:**

Neural-net based image compression is a field which is about to get hot, and this paper asks the obvious question: can we design a neural-net based image compression algorithm such that the features it produces are useful for classification & segmentation?

The fact that it's an obvious question does not mean that it's a question that's worthless. In fact, I am glad someone asked this question and tried to answer it.

Pros:
- Clear presentation, easy to follow.
- Very interesting, but obvious, question is explored.
- The paper is very clear, and uses building blocks which have been analyzed before, which leaves the authors free to explore their interactions rather than each individual building block's property.
- Results are shown on two tasks (classification / segmentation) rather than just one (the obvious one would have been to only discuss results on classification), and relatively intuitive results are shown (i.e., more bits = better performance). What is perhaps not obvious is how much impact does doubling the bandwidth have (i.e., initially it means more, then later on it plateaus, but much earlier than expected).
- Joint training of compression + other tasks. As far as I know this is the first paper to talk about this particular scenario.
- I like the fact that classical codecs were not completely discarded (there's a comparison with JPEG 2K).
- The discussion section is of particular interest, discussing openly the pros/cons of the method (I wish more papers would be as straightforward as this one).

Cons:
- I would have liked to have a discussion on the effect of the encoder network. Only one architecture/variant was used.
- For PSNR, SSIM and MS-SSIM I would like a bit more clarity whether these were done channel-wise, or on the grayscale channel.
- While runtime is given as pro, it would be nice for those not familiar with the methods to provide some runtime numbers (i.e., breakdown how much time does it take to encode and how much time does it take to classify or segment, but in seconds, not flops). For example, Figure 6 could be augmented with actual runtime in seconds.
- I wish the authors did a ctrl+F for "??" and fixed all the occurrences.
- One of the things that would be cool to add later on but I wished to have beeyn covered is whether it's possible to learn not only to compress, but also downscale. In particular, the input to ResNet et al for classification is fixed sized, so the question is -- would it be possible to produced a compact representation to be used for classification given arbitrary image resolutions, and if yes, would it have any benefit?

General comments:
- The classification bits are all open source, which is very good. However, there are very few neural net compression methods which are open sourced. Would you be inclined to open source the code for your implementation? It would be a great service to the community if yes (and I realize that it could already be open sourced -- feel free to not answer if it may lead to break anonymity, but please take this into consideration).

---

> ### Author Response · Authors · 2017-12-23
> **Response to AnonReviewer2**
>
> Thank you for your positive feedback. As for your comments, we have addressed them as follows:
> 1) The reason for not exploring more encoders is twofold. First, few of the state-of-the-art compression variants of neural networks are open-source which makes the implementation of different architectures very time consuming and difficult. We picked this particular encoder since it has been used in at least 2 different works, Theis et al. and Agustsson et al. Second, the autoencoder compression methods also all have a similar structure and thus one could expect the performance to be similar for different encoders. This is nevertheless an interesting direction of research worth pursuing.
> 2) These were done channel-wise (RGB).
> 3) We have not yet had time to do this as the code for segmentation, classification and compression networks have different levels of optimization (e.g., NCHW vs NHWC tensor layout), so doing a fair comparison is time consuming and involves careful engineering. We however can do this before camera ready, if the paper is accepted. We also note that the architectures used for compression and inference are very similar, convolutional networks with residual blocks, and therefore FLOPs should be a good proxy metric.
> 4) This has been fixed in the revised edition of the paper
> 5) We agree that it would be interesting to learn the downscaling as well, after all it is just an anti-aliasing kernel (i.e. a convolution with a fixed kernel) followed by a subsampling operation and can be learned as well.  However, processing the images in full-resolution during training would quite significantly increase training times, which are already pushing our limits in terms of compute resources. Another challenge is that hyperparameters of the ResNet architecture would likely need to be re-tuned. We chose to adhere to the standard setting of the classification literature, so that we could use the hyperparameter settings and training schedules which have already been optimized extensively.

---

### Official Review · AnonReviewer1 · 2017-11-27
**Review for "Towards Image Understanding from Deep Compression Without Decoding"**

**Rating:** 6
**Confidence:** 4

**Review:**

Thanks for addressing most of the issues. I changed my given score from 3 to 6.

Summary:
This work explores the use of learned compressed image representation for solving 2 computer vision tasks without employing a decoding step.

The paper claims to be more computationally and memory efficient compared to the use of original or the decompressed images. Results are presented on 2 datasets "Imagenet" and "PASCAL VOC 2012". They also jointly train the compression and classification together and empirically shows it can improve both classification and compression together.

Pros:
+ The idea of learning from a compressed representation is a very interesting and beneficial idea for large-scale image understanding tasks.

Cons:
- The paper is too long (13 pages + 2 pages of references). The suggested standard number of pages is 8 pages + 1 page of references. There are many parts that are unnecessary in the paper and can be summarized. Summarizing and rewording them makes the paper more consistent and easier to read:
  ( 1. A very long introduction about the benefits of inferring from the compressed images and examples.
    2. A large part of the intro and Related work can get merged.
    3. Experimental setup part is long but not well-explained and is not self-contained particularly for the evaluation metrics.
    “Please briefly explain what MS-SSIM, SSIM, and PSNR stand for”. There is a reference to the Agustsson et al 2017 paper
     “scalar quantization”, which is not well explained in the paper. It is better to remove this part if it is not an important part or just briefly but clearly explain it.
     4. Fig. 4 is not necessary. 4.3 contains extra information and could be summarized in a more consistent way.
     5. Hyperparameters that are applied can be summarized in a small table or just explain the difference between the
      architectures that are used.)

- There are parts of the papers which are confusing or not well-written. It is better to keep the sentences short and consistent:
E.g: subsection 3.2, page 5: “To adapt the ResNet … where k is the number of … layers of the network” can be changed to 3 shorter sentences, which is easier to follow.
There are some typos: e.g: part 3.1, fever ---> fewer,

- As it is mentioned in the paper, solving a Vision problem directly from a compressed image, is not a novel method (e.g: DCT coefficients were used for both vision and audio data to solve a task without any decompression). However, applying a deep representation for the compression and then directly solving a vision task (classification and segmentation) can be considered as a novel idea.

- In the last part of the paper, both compression and classification parts are jointly trained, and it is empirically presented that both results improved by jointly training them. However, to me, it is not clear if the trained compression model on this specific dataset and for the task of classification can work well for other datasets or other tasks.
The experimental setup and the figures are not well explained and well written.

---

> ### Author Response · Authors · 2017-12-23
> **Response to AnonReviewer1**
>
> Thank you for your your review, we have considered your comments in the revised version of the paper. Given the improved paper and positive perspective of the other reviews, we hope you reconsider your rating.
>
> For specific points:
>
> Regarding paper length: since this is a study paper, we felt it benefited from verbosity. However, we have managed to shorten the paper to 9.5 pages, while keeping the original story intact. We followed most of your suggestions: (1-2) We shortened the introduction and related work; (4) We made Section 4.3 (now Section 4.4) much more concise and moved Figure 4 to the appendix as it did not contain core results of our work. (3) We added a better description of the compression metrics to the experiments section. However, we also moved the compression results to the appendix, and added a more detailed explanation of the metrics there. (5) We also fixed wording in the paper as you suggested and moved hyperparameter settings and details to the appendix, as we felt these details distract from the main message of the paper. In addition to this, we refined presentation of joint training results using plots rather then presenting them in text.
>
> As we mention in the paper, learning from DCT (of JPEG) has been done before. However, our setting of using features from learned compression networks is significantly different. The DCT of JPEG is simply a linear transform over 8x8 patches, whereas the compressed representation is a feature map from a deep convolutional neural network. This opens directions such as joint learning of compression and inference (see Section 6) and warrants a full study of the problem.
>
> To show that the improvement of joint training generalizes to another task, we added an experiment: We take the (jointly trained) classification network and finetune it for segmentation. The results are shown in Figure 7, where the resulting network significantly outperforms the separately trained network - achieving a significant performance boost of 1.1-1.8% higher mIoU depending on the compression operating point. See Figure 7 and discussion in Section 6.2 of the revised paper for more details.
> We emphasize that this generalization is also occurring across datasets, from ILSVRC2012 (classification) to PASCAL VOC (segmentation).
>
> Finally, we made an effort to better clarify and describe the experiments.

---

### Official Review · AnonReviewer3 · 2017-11-27
**Training for Compression, Classification and Segmentation**

**Rating:** 6
**Confidence:** 3

**Review:**

This is a well-written and quite clear work about how a previous work on image compression using deep neural networks can be extended to train representations which are also valid for semantic understanding. IN particular, the authors tackle the classic and well-known problems of image classification and segmentation.

The work evolves around defining a loss function which initially considers only a trade-off between reconstruction error and total bit-rate. The representations trained with the loss function, at three different operational points, are used as inputs for variations of ResNet (image classification) and DeepLab (segmentation). The results obtained are similar to a ResNet trained directly over the RGB images, and actually with a slight increase of performance in segmentation. The most interesting part is a joint training for both compression and image classification.

PROS
P.1 Joint training for both compression and classification. First time to the authors knowledge.
P.2 Performance on classification and segmentation tasks are very similar when compared to the non-compressed case with state-of-the-art ResNet architectures.
P.3 Text is very clear.
P.4 Experimentation is exhaustive and well-reported.

CONS
C1. The authors fail into providing a better background regarding the metrics MS-SSIM and SSIM (and PSNR, as well) and their relation to the MSE used for training the network. Also, I missed an explanation about whether high or low values for them are beneficial, as actually results compared to JPEG and JPEG-2000 differ depending on the experiment.
C2. The main problem is of the work is that, while the whole argument is that in an indexing system it would not be necessary to decompress the representation coded with a DNN, in terms of computation JPEG2000 (and probably JPEG) are much lighter that coding with DNN, even if considering both the compression and decompression. The authors already point at another work where they explore the efficient compression with GPUs, but this point is the weakest one for the adoption of the proposed scheme.
C3. The paper exceeds the recommendation of 8 pages and expands up to 13 pages, plus references. An effort of compression would be advisable, moving some of the non-core results to the appendixes.

QUESTIONS
Q1. Do you have any explanation for the big jumps on the plots of Figure 5 ?
Q2. Did you try a joint training for the segmentation task as well ?
Q3. Why are the dots connected in Figure 10, but not in Figure 11.
Q4. Actually results in Figure 10 do not seem good... or maybe I am not understanding them properly. This is related to C1.
Q5. There is a broken reference in Section 5.3. Please fix.

---

> ### Author Response · Authors · 2017-12-23
> **Response to AnonReviewer3**
>
> Thank you for your review. We considered your comments as follows.
> C1 We have added definitions and clarification on the metrics to the paper. For all of them high means good.
> C2 We  gave an indexing system as an example application, but the main goal of the paper is to study the interplay between learned compression and inference in general. As mentioned in the paper, classical compression can be much faster than learned one - and our computational gains are relative to a system using learned compression. While falling back to classical codecs would be cheaper in terms of compute (since classical encoder+decoder is more efficient than the learned encoder), the story is not so simple, since this would come at the expense of transmitting more data for a given target image quality.  This can be crucial, since for mobile devices, data transmission (I/O) is responsible for most energy usage in common applications  (Pathak et. al, 2012). Since learned compression is still at its infancy, we expect the gap in terms of compression performance between learned methods and classical ones to grow. Furthermore, with the increasing availability of dedicated neural network processing units on devices, deep image compression methods could become as fast as traditional ones.
> C3 We have condensed the paper to remove redundant text and also moved some non-core results to the appendix.
>
> Q1. Yes, the standard learning rate schedule for training the ResNet classification architectures is a constant learning rate divided by factor 10 at fixed points in the training (every 8 epochs for our setting). At the point when the learning rate is lowered the validation accuracy increases rapidly, and our validation curves show these jumps clearly.
> The jumps/difference between operating points is due to more detail being present in images at higher bitrate (higher bpp) and therefore doing inference on them is easier.
> Q2. Yes we also did experiments for joint training with segmentation that are detailed in the revised version of the paper. In short, we do not train jointly on the segmentation task but we take the jointly trained classification network (that improves classification) and use that as a starting point for segmentation, showing significant improvement for segmentation. These results are shown in Figure 7.
> Q3. We have made the style of the plots consistent, connecting the dots for both.
> Q4. Figure 10 (also Figure 10 in the revised edition) shows how the compression metrics change when training jointly compared to training only the compression network. It can be seen that training jointly improves the perceptual metrics MS-SSIM and SSIM slightly while PSNR gets slightly worse (higher is better for all metrics). Figure 10’s main point is that the image compression metrics do not get worse in two out of three metrics as we do joint training. At the same time, Figure 7 shows that the inference  performance (both segmentation and classification) significantly improves. See Section 6.2 for a thorough discussion in the revised edition. As this is not a core result it was moved to the appendix.
> Q5. We have fixed this in the revised edition of the paper.
>
> (Pathak, A., Hu, Y. C., & Zhang, M. (2012, April). Where is the energy spent inside my app?: fine grained energy accounting on smartphones with eprof. In Proceedings of the 7th ACM european conference on Computer Systems (pp. 29-42). ACM.)

---

### Decision · Program_Chairs · 2018-01-29
**ICLR 2018 Conference Acceptance Decision**

**Decision:**

Accept (Poster)

**Comment:**

Some reviewers seem to assign novelty to the compression and classification formulation; however, semi-supervised autoencoders have been used for a long time. Taking the compression task more seriously as is done in this paper is less explored.

The paper provides some extensive experimental evaluation and was edited to make the paper more concise at the request of reviewers. One reviewer had a particularly strong positive rating, due to the quality of the presentation, experiments and discussion. I think the community would like this work and it should be accepted.